# Assessing the availability and scope of routine data on post-pregnancy family planning: A cross-sectional review of registers and reporting tools in 18 low- and middle-income countries

**Deborah Sitrin**[1]*, **Aurélie Brunie**[2], **Rebecca Rosenberg**[3], **Lucy Wilson**[4], **Elena Lebetkin**[5], **Rogers Kagimu**[6], **Fredrick Makumbi**[7]

1 Jhpiego, Washington, District of Columbia, United States of America, 2 FHI 360, Washington, District of Columbia, United States of America, 3 Avenir Health, Takoma Park, Maryland, United States of America, 4 Rising Outcomes, Hillsborough, North Carolina, United States of America, 5 FHI 360, Durham, North Carolina, United States of America, 6 Division of Health Information Management and Division of Reproductive and Infant Health, Ministry of Health, Kampala, Uganda, 7 Department of Epidemiology and Biostatistics, School of Public Health, College of Health Sciences, Makerere University, Kampala, Uganda

* dsitrin1@jh.edu

## Abstract

Many low- and middle-income countries (LMICs) include postpartum and postabortion family planning (PPFP/PAFP) in their national family planning (FP) commitments. Understanding what PPFP and PAFP data are available in routine health information systems (HIS) is important, as both county-level and global monitoring increasingly rely on these systems to track service delivery and scale-up, inform program improvements, and support accountability. This paper reviews the availability of PPFP and PAFP data elements in HIS across 18 LMICs. We analyzed 85 facility registers and 31 monthly summary forms covering antenatal care (ANC), labor and delivery (L&D), postnatal care (PNC), FP, and postabortion care (PAC). All 18 countries record PPFP provision in registers and summary forms; 14 also capture PAFP provision in registers, with 10 reporting it in summary forms. Most (15/18) collect immediate PPFP (≤48 hours after childbirth), in alignment with recommendations from the PPFP Community of Practice and High Impact Practices partnership, though 6 need to add this to their summary forms to improve data accessibility. Fourteen countries collect PPFP at multiple time points (e.g., ≤48 hours and ≤6 weeks). While all collect client age in registers, only one disaggregates PPFP and two disaggregate PAFP by age in summary forms. There is variation in the contraceptive methods recorded and compiled. Documentation of FP counseling is less consistent: 8 countries record it during ANC (2 in summary forms), 7 before discharge after childbirth (2 in summary forms), and 10 during PNC (2 in summary forms). Differences in timing, disaggregation, and method detail affect cross-country comparability, though several countries collect

**Data availability statement:** Summaries of analyzed tools available on Figshare: https://doi.org/10.6084/m9.figshare.28904894.v2.

**Funding:** This work was supported by the US Agency for International Development through the MOMENTUM Country and Global Leadership project led by Jhpiego (Cooperative Agreement #7200AA20CA00002) and the Research for Scalable Solutions project led by FHI 360 (Cooperative Agreement #7200AA19CA00041), and by the Bill & Melinda Gates Foundation (INV-040052) prior to the conclusion of these awards in early 2025. The contents are the responsibility of authors and do not necessarily reflect the views of the United States Government or the Foundation. Under the grant conditions of the Foundation, a Creative Commons Attribution 4.0 License has already been assigned to the Author Accepted Manuscript version that might arise from this submission.

**Competing interests:** The authors have declared that no competing interests exist.

sufficiently aligned data for meaningful analysis. Country efforts to track PPFP across multiple contact points suggest a commitment to broad integration, which should be matched by expanded global indicator guidance that reflects the full scope of service delivery across the continuum of care.

## Introduction

The majority of postpartum women in low- and middle-income countries (LMICs) wish to delay pregnancy for at least one year [1], yet unmet need for family planning (FP) among postpartum women is high, often exceeding that of non-postpartum women [2–5]. Extensive evidence confirms that postpartum contraception is safe, effective, and cost-efficient. In 2013, the World Health Organization (WHO) endorsed integrating FP into pregnancy, childbirth, postnatal, and child care in its *Programming Strategies for Postpartum Family Planning* (PPFP) [6]. Immediate postpartum family planning (IPPFP) – the provision of contraception counseling and services as part of facility-based childbirth care – was recognized by the High Impact Practices (HIP) partnership as a "proven" intervention that increases contraceptive uptake [7]. Despite strong global guidance supporting PPFP and clear evidence of benefit, gaps in access and use persist across LMICs.

Post-abortion care (PAC) also presents a critical touchpoint for contraceptive counseling and provision. Many women who receive PAC are highly motivated to use contraception, are already engaged with health care providers, and experience a rapid return to fertility [8]. Like PPFP, post-abortion family planning (PAFP) is safe, efficient, and recognized as a "proven" HIP [9]. Yet, many PAC clients are discharged without any form of contraception [9]. While global initiatives such as FP2020 and FP2030 have driven national commitments to expand PPFP and PAFP services, progress in implementation has been uneven. Many countries remain in the early stages of implementation, and translating commitments into widespread access and use continues to be a challenge [10].

Routine data are essential for planning, monitoring, and improving healthcare services [11]. In LMICs, public facilities typically record patient services in paper registers maintained by individual service units, which are then compiled into monthly summary reports [12]. These summaries are entered into an electronic platform by facility or district-level staff, making the data available to managers and policymakers at higher levels of the health system [13]. A 2018 USAID-funded review of data tools in 18 countries found that only half reported PPFP clients and just 2 reported PAFP clients; however, the report did not include sufficient detail to understand exactly what the indicators captured, such as the timing of PPFP services, the contraceptive methods recorded, or whether data were disaggregated by age [14].

The scarcity of routine data prompted the global PPFP Community of Practice to establish a measurement committee to develop consensus recommendations for key PPFP indicators [15]. These were finalized in 2019 (Box 1), and two recommended indicators were later included in the HIP brief for countries implementing IPPFP, with

two similar indicators featured in the PAFP brief (Table 1). In 2023, global partners issued a renewed *Call to Action* to scale up PPFP and PAFP as integral components of universal health coverage and primary health care, which included a focus on improving routine Health Information System (HIS) indicators for voluntary contraception counseling and uptake [16].

> ### Box 1. PPFP Community of Practice indicator recommendations.
>
> 1. All countries include this indicator for PPFP uptake in their national health information system: Percent of women who deliver in a facility and initiate or leave with a modern contraceptive method prior to discharge. (Recommended to be collected in the Delivery Care/Maternity register *or* the Postnatal Care register, if used for pre-discharge care. Recommended to capture women initiating any modern contraceptive method, including the lactational amenorrhea method (LAM), and for reports to disaggregate the total number by method and age group in countries where adolescent birth rates are high.)
>
> 2. Countries may also include this additional indicator on FP counseling prior to discharge: Percent of women who deliver in a facility and received counseling on FP prior to discharge.
>
> 3. Document PPFP counseling and method choice during pregnancy. Documenting if FP counseling is done during antenatal care reminds providers to start counseling early, but the data do not need to be aggregated given the burden it can place on facility staff using paper forms.
>
> 4. Additional testing is needed to develop an indicator for PPFP uptake during the extended postpartum period. (The committee agreed an indicator of PPFP beyond the immediate period was needed, but there was not sufficient experience with a single indicator that would be useful across all countries.)

To support this momentum, FP2030 convened the *Accelerating Access to Postpartum and Postabortion Family Planning Workshop* in 2023 with delegates from 15 countries and global stakeholders to advance national PPFP and PAFP action plans aligned with their FP2030 commitments [17]. A key focus of the meeting was strengthening data and measurement to support evidence-based decision-making. In preparation, the USAID-funded MOMENTUM Country and Global Leadership program reviewed current HIS tools from participating countries, most of which had been revised since the 2018 review, with the exception of Mozambique and Tanzania. This updated review provided an opportunity to assess not only whether PPFP and PAFP data elements are being tracked, but also how they are defined and whether they align with global recommendations introduced after 2018. The Research for Scalable Solutions (R4S) and Supporting Measurement and Replicable Techniques for HIPs (SMART-HIPs) projects expanded this review to include three additional countries in early 2025.

**Table 1. Recommended indicators from high impact practices briefs.**

| HIP | HIP Recommended Indicators | HIP Recommended Disaggregation |
|---|---|---|
| IPPFP | Number/Percent of women who delivered in a facility and received counseling on FP prior to discharge | Age (<20 & ≥ 20) |
| | Number/Percent of women who deliver in a facility and initiate or leave with a modern contraceptive method prior to discharge | Type of method Age (<20 & ≥ 20) |
| PAFP | Percent of postabortion clients who were counseled on FP | Age (<20 & ≥ 20) |
| | Percent of postabortion clients who leave the facility with a modern contraceptive | Type of method Age (<20 & ≥ 20) |

Source: HIP briefs for Immediate postpartum family planning [7] and postabortion family planning [9]

This paper presents consolidated findings from the updated review across 18 countries, highlighting the state of PPFP and PAFP data availability to support national efforts to accelerate progress and strengthen accountability. These findings provide a practical reference for countries—whether included in the review or not—to evaluate whether their routine HIS are equipped to monitor the availability and uptake of these high-impact services. The review also examines the degree of alignment between national data elements and global indicator recommendations developed by the PPFP Community of Practice and featured in HIP briefs. Strengthening indicator alignment and data quality can improve data interpretation, support evidence-based decision-making, enable cross-country learning, and contribute to tracking progress toward global FP and health goals [18]. As routine data systems continue to mature and reliance on population-based surveys becomes more uncertain due to cost and frequency concerns, robust HIS data will be increasingly essential for monitoring and advancing equitable access to voluntary PPFP and PAFP services [19–21].

## Methods

Our approach was informed by past reviews [14,18]. We conducted a cross-sectional review of the most recent version of facility registers used by healthcare providers to record services given to each patient and the monthly summary form used by facilities to report the total number of services provided to higher levels of the health system. We sought tools that were approved by the Ministry of Health for use in public facilities. Of the 18 countries included in this review, 15 were included in the initial review conducted by the authors from the MOMENTUM program and 5 in the review independently conducted by the authors from the R4S and SMART-HIPs projects (Burkina Faso, Mozambique, Nepal, Nigeria, and Uganda); two countries (Nepal and Nigeria) were included in both reviews.

### Gathering tools

Because PPFP services can be offered throughout the continuum of care from pregnancy to one year after childbirth, we aimed to collect five types of registers from each county: antenatal care (ANC), labor and delivery (L&D) or maternity, postnatal care (PNC), FP, and PAC (Fig 1). In Fig 1, the PNC register overlaps with L&D because in some countries, PNC registers are used both for services provided before discharge following childbirth and for follow-up visits, including for women who delivered elsewhere. While PPFP data may also be recorded in individual patient files kept at facilities or in patient-held cards, such data are not compiled in monthly forms and were therefore excluded from this review. We gathered copies of paper-based tools by searching project archives and requesting current versions from country partners. Tools were requested based on what was actively in use within health facilities; however, the exact year of finalization was not always identifiable. Formats included PDFs, photographs of printed tools, or Excel files. Prior to the FP2030 Nepal workshop, collected tools

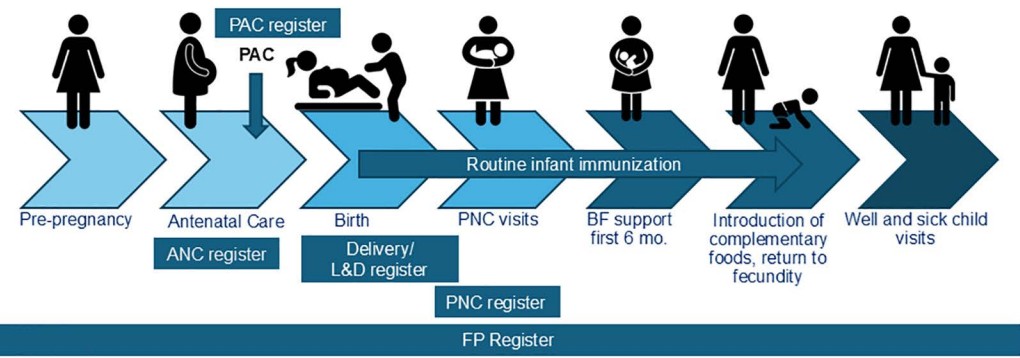

ANC=Antenatal Care; BF=Breastfeeding; FP=Family Planning; L&D=Labor and Delivery; PAC=Postabortion care; PNC=Postnatal care

**Fig 1. Types of registers reviewed.**

were uploaded onto a shared Google Drive, and country delegations were invited to verify that the most up-to-date tools were included. Not all registers depicted in Fig 1 exist in every country and some are combined. For example, PAC or PNC services may be recorded within the L&D register, or ANC and PNC may be documented in a single combined register.

Some countries included in this review (Bangladesh, Pakistan, and Indonesia) have more than one HIS managed by different government departments. In each country, we requested tools from the department primarily responsible for collecting PPFP data. For Bangladesh, we collected tools used by facilities managed by the Directorate General of Family Planning but did not obtain tools from the Directorate General of Health Services, which oversees FP services in hospitals and other facilities under its jurisdiction. In Pakistan's devolved health system, each province operates two separate HIS. We obtained Department of Health tools from 4 provinces (Balochistan, Khyber Pakhtunkhwa (KP), Punjab, and Sindh) but did not include tools from the provincial Population Welfare Departments, which manage non-facility-based FP services. For Indonesia, we obtained the form used to document FP services in the electronic Family Information System (SIGA), managed by the Ministry of Population and Family Development; however, tools from the Ministry of Health were not obtained.

### Capturing data elements, analysis, and verification

Tools (registers and summary forms) in other languages were translated into English using Google Translate or DeepL. MOMENTUM and R4S/SMART-HIPs used separate Microsoft Excel worksheets for data abstraction. MOMENTUM created a new row in its worksheet for each instance where one of the following data elements was found in a tool: PPFP counseling during ANC, before discharge, during PNC, at another time, or when timing was unspecified; PPFP method provision before discharge/within 48 hours, within or at 6 weeks, within 1 year, at another time, or when timing was unspecified; PAFP counseling; and PAFP method provision. For each data element identified, columns in the worksheet documented the country, the name of the register or form, age groups recorded or aggregated, and FP methods recorded or aggregated. For example, if a register allowed providers to indicate whether FP was given within 48 hours, within or at 6 weeks, or up to a year after birth, three separate rows were created. An additional column captured notes about how PPFP or PAFP data were to be entered, as described in the tool, and any deviations from the defined data element. For instance, if a PNC register included PPFP without specifying timing, it was assumed to refer to provision within or at 6 weeks, and this assumption was noted in the worksheet. R4S/SMART-HIPs used a different approach, creating one row per country and separate columns for each data element, including notes within the same column.

For both projects, two authors independently abstracted information from all tools, then reviewed their findings together to identify and resolve discrepancies. Preliminary findings were presented in tables, which were validated by country delegates during the FP2030 Nepal workshop for the MOMENTUM project. R4S/SMART-HIPs reviewers conducted follow-up calls with Ministry of Health (MOH) personnel or other individuals familiar with the HIS in each of the 5 countries included in their review. In addition, authors from MOMENTUM and R4S/SMART-HIPs met to discuss and resolve a few discrepancies in the initial results from the two overlapping countries (Nepal and Nigeria). Final results were combined in tables with one row for each country (plus 4 rows for each Pakistan province, with a row for the country to document if a data element is collected by at least 1 province). Separate tables were created to examine PPFP and PAFP counseling, PPFP and PAFP provision timing, PPFP method, and PAFP method.

### Ethics

This work was determined to be non-human subject research by the Johns Hopkins Bloomberg School of Public Health Institutional Review Board and the FHI 360 Office of Human Research Ethics.

### Results

We reviewed 85 registers and 31 summary forms for data elements on PPFP and PAFP across the 18 countries. In some countries, separate method-specific FP registers or unit-specific monthly summary forms contributed to a higher number

of tools, while other countries used combined registers, such those that document both ANC and PNC (Table 2). We distinguished between PAC registers known or presumed not to exist (-) versus tools we were unable to obtain (Missing). Tables 3–6 summarize the contents of registers and monthly summary forms with a check (✓) indicating at least one register in the country has space to record that data element but it is not on the monthly summary form and an envelope (✉) indicating the data element is included in the monthly summary form and thus reported to higher levels of the health system.

### Postpartum family planning

Starting with the HIP-recommended counseling indicator, registers in 7 countries capture women counseled on FP before discharge after childbirth, and summary forms in two of these countries (Sierra Leone and Tanzania) include this data element (Table 3). Additionally, registers in 8 countries capture FP counseling during ANC, registers in 10 countries capture FP counseling during postnatal care or within 6 weeks after childbirth, and registers in 4 countries record PPFP counseling without specifying when counseling was provided. Each of these data elements is included in summary forms in 2 countries. Overall, 14 of the 18 countries record at least one data element on PPFP counseling in registers, and 7 countries include at least one data element on PPFP counseling in summary forms. While 10 countries record counseling at more than one contact point in registers, only Nigeria includes two data elements on PPFP counseling in summary forms (counseling during ANC and PPFP counseling with unspecified timing).

**Table 2. Registers and summary forms reviewed.**

| Country | Language | FP Register | ANC Register | L&D Register | PNC Register | PAC Register | Monthly summary form |
|---|---|---|---|---|---|---|---|
| Bangladesh (DGFP) | Bangla | Date unknown[1] | Date unknown[2] | 2016 | Date unknown[2] | – | 2018 |
| Burkina Faso | French | Date unknown | Date unknown | Date unknown | Date unknown | Date unknown | Date unknown |
| Ethiopia | English | 2021 | 2021 | 2021 | 2021 | 2021 | 2017 |
| Ghana | English | 2022 | 2022 | 2022 | 2022 | Date unknown | Date unknown[1] |
| India | English | 2016[1] | 2013 | 2013 | 2013 | Date unknown | Date unknown |
| Indonesia (SIGA) | Indonesian | Date unknown | – | – | – | – | Date unknown |
| Kenya | English | 2019 | 2019 | 2019 | 2019 | – | 2019 |
| Malawi | English | 2023 | 2023 | 2017 | 2017 | Date unknown | Various dates[1] |
| Mozambique | Portuguese | 2015 | 2015 | 2015 | 2015 | 2015 | 2015[1] |
| Nepal | Nepali | 2022/2023 | 2022/2023[2] | 2022/2023[2] | 2022/2023[2] | 2022/2023 | 2022/2023 |
| Nigeria | Engish | 2019 | 2019 | 2019 | 2019 | – | 2019 |
| Pakistan (Balochistan) | English | Missing | Date unknown[2] | Date unknown | Date unknown[2] | – | Date unknown |
| Pakistan (KP) | English | Date unknown | Date unknown[2] | Date unknown | Date unknown[2] | – | Missing |
| Pakistan (Punjab) | English | Date unknown | Date unknown[2] | Date unknown | Date unknown[2] | – | Date unknown |
| Pakistan (Sindh) | English | Date unknown | Date unknown[2] | Date unknown | Date unknown[2] | – | Missing |
| Philippines | English | Date unknown | Date unknown[2] | Date unknown[2] | Date unknown | – | Date unknown |
| Rwanda | French (PNC)[3] | 2017 | 2022 | 2022 | 2017 | – | 2018 |
| Sierra Leone | English | Date unknown | Date unknown[2] | Date unknown | Date unknown[2] | – | Date unknown |
| Tanzania | Swahili | 2017 | 2017 | 2017 | 2017 | – | 2017[1] |
| Uganda | English | 2024 | 2024 | 2024 | 2024 | – | 2024 |
| Zambia | English | 2023 | 2023 | 2023 | 2023 | – | 2023 |

ANC = Antenatal Care; FP = Family Planning; L&D = Labor and Delivery; PAC = Postabortion care; PNC = Postnatal care.

[1]Multiple registers or reporting forms (e.g., register for each contraceptive method or reporting form for each unit)

[2]Combined ANC, L&D, and/or PNC registers

[3]PNC register in French only and translated by authors into English. All other registers/form in English and French

**Table 3. Data elements on PPFP and PAFP counseling across 18 countries.**

| Country | PPFP counseling: | | | | | PAFP counseling¹ |
|---|---|---|---|---|---|---|
| | During ANC | Before discharge after birth¹ | During PNC/≤ 6 weeks | Other timing | Timing unspecified | |
| Bangladesh DGFP | – | – | – | – | – | – |
| Burkina Faso | ✓ | ✓ | ✓ | – | – | – |
| Ethiopia | ✓ | – | ✓ | – | – | ✓ |
| Ghana | – | ✓ | ✓ | – | – | ⊠ |
| India | ✓ | ✓ | ✓ | – | – | – |
| Indonesia SIGA | – | – | – | – | – | – |
| Kenya | ✓ | – | ⊠ | – | – | – |
| Malawi | – | – | ⊠ | – | – | ✓ |
| Mozambique | – | – | – | – | – | – |
| Nepal | – | – | – | – | – | – |
| Nigeria | ⊠ | ✓ | ✓ | – | ⊠ | ✓ |
| Pakistan (≥1 province) | – | – | – | – | ⊠ | ⊠ |
| Balochistan | – | – | – | – | ⊠ | ⊠ |
| KP | – | – | – | – | ✓² | ✓² |
| Punjab | – | – | – | – | – | – |
| Sindh | – | – | – | – | – | ✓² |
| Philippines | ✓ | – | ✓ | – | – | – |
| Rwanda | ⊠ | – | – | – | – | – |
| Sierra Leone | – | ⊠ | – | – | – | – |
| Tanzania | – | ⊠ | ✓ | – | – | – |
| Uganda | ✓ | ✓ | – | – | ✓ | ✓ |
| Zambia | – | – | ✓ | ✓ | ✓ | ✓ |
| No. of countries with data element in registers | 8 | 7 | 10 | 1 | 4 | 7 |
| No. of countries with *any* data element in registers | 14 | | | | | |
| No. of countries with data element in summary form | 2 | 2 | 2 | 0 | 2 | 2 |
| No. of countries with *any* data element in summary form | 7 | | | | | |

¹HIP recommended

²Missing summary form so unable to assess if reported

Looking at the HIP-recommended PPFP provision indicator, registers in 15 of 18 countries capture contraceptive method provision before discharge to women who gave birth, and summary forms in 9 countries include this data element (Table 4). Registers in 12 countries capture method provision within 6 weeks of childbirth (6 in summary forms), 8 countries capture method provision within one year (3 in summary forms), 7 countries capture a different timeframe (4 in summary forms), and 10 countries an unspecified timeframe (8 in summary forms). Other time-frames include 6–7 days (Ghana, India, Malawi, Zambia); Tanzania breaks down the first 6 weeks to 48 hours, 2–3 days, 8–28 days, and 29–42 days; Burkina Faso uses ≤48 hours or 48 hours- 24 months; Uganda uses within 48 hours, 48 hours- 4 weeks, 4–6 weeks, 6 weeks – 6 months, and 6–12 months. Fourteen countries capture method provision at more than one timeframe in registers; 9 of which include more than one timeframe in their summary forms as well. All 18 countries include at least one data element related to postpartum method provision in a register and summary form.

Table 4. Data elements on PPFP and PAFP method provision across 18 countries.

| Country | PPFP method provision | | | | | PAFP method provision[1] |
|---|---|---|---|---|---|---|
| | Before discharge/ ≤48 hrs[1] | During PNC/ ≤6 weeks | Within 1 year | Other timing | Timing unspecified | |
| Bangladesh DGFP | ✓ | – | – | – | ⊠ | – |
| Burkina Faso | ⊠ | ✓ | – | ⊠ | – | ⊠ |
| Ethiopia | ⊠ | – | – | – | – | ✓ |
| Ghana | ✓ | ✓ | ✓ | ✓ | ⊠ | ⊠ |
| India | ⊠ | ✓ | – | ⊠ | ✓ | ⊠ |
| Indonesia SIGA | – | – | – | – | ⊠ | ⊠ |
| Kenya | ⊠ | ⊠ | ✓ | – | ⊠ | ⊠ |
| Malawi | – | ⊠ | – | ⊠ | – | ⊠ |
| Mozambique | ⊠ | – | – | – | – | ✓ |
| Nepal | ⊠ | ✓ | ⊠ | – | – | ⊠ |
| Nigeria | ✓ | – | – | – | ⊠ | ✓ |
| Pakistan (≥1 province) | ✓[2] | ✓[2] | ✓[2] | – | ⊠ | ⊠ |
| Balochistan | – | – | – | – | ⊠ | ⊠ |
| KP | – | – | – | – | ✓[2] | ✓[2] |
| Punjab | – | – | – | – | ⊠ | – |
| Sindh | ✓[2] | ✓[2] | ✓[2] | | | ✓[2] |
| Philippines | – | – | – | | ⊠ | – |
| Rwanda | ⊠ | ⊠ | – | – | – | – |
| Sierra Leone | ⊠ | ✓ | ✓ | – | ⊠ | – |
| Tanzania | ✓ | ⊠ | ⊠ | ✓ | – | ⊠ |
| Uganda | ⊠ | ⊠ | ⊠ | ⊠ | – | ⊠ |
| Zambia | ✓ | ⊠ | ✓ | ✓ | ✓ | ✓ |
| No. of countries with data element in registers | 15 18 | 12 | 8 | 7 | 10 | 14 14 |
| No. of countries with *multiple* data elements in registers | 14 | | | | | Not applicable |
| No. of countries with data element in summary form | 9 9 | 6 | 3 | 4 | 8 | 10 |
| No. of countries with *multiple* data elements in summary form | | | | | | Not applicable |

[1]HIP recommended

[2]Missing summary form so unable to assess if reported

While all 18 countries collect client age in the registers used to record PPFP, only Ethiopia disaggregates PPFP provision by age in its summary form, using four categories: 10–14, 15–19, 20–24, and 25 years old or older (results not shown). There is greater variation in the contraceptive methods captured and reported (Table 5). Emergency contraceptive (EC) pills and LAM are rarely recorded with only 5 and 7 countries, respectively, capture these in their registers. More countries include intrauterine devices (IUDs), implants, sterilization, pills, condoms, and injectables. Notable exceptions include Bangladesh where only postpartum IUDs, implants, or sterilization are captured (other methods are not counted); Mozambique where only sterilization is recorded; India where IUD, sterilization, and injectables are captured, but not implants, pills, or condoms; and Ethiopia where IUD, implant, sterilization, and pills are captured but not condoms or injectables (likely because Ethiopia only tracks immediate PPFP). In Mozambique, India, and Ethiopia, clients receiving any other method are grouped under "other." Only 4 countries (Burkina Faso, Kenya, Malawi, Zambia) capture all methods listed in Table 4 in their registers. Registers

**Table 5. Capturing and disaggregating PPFP method across 18 countries.**

| Country | IUD | Implant | Steril-ization | Pill | Con-dom | Inject-able[1] | EC | LAM | Other[3] | # of methods (Registers) | # of methods (Summary forms) |
|---|---|---|---|---|---|---|---|---|---|---|---|
| Bangladesh DGFP | ⊠ | ✓ | ⊠ | – | – | – | – | – | – | 3 | 2 |
| Burkina Faso[2] | ⊠ | ⊠ | ⊠ | ⊠ | ⊠ | ⊠ | ⊠ | ⊠ | ⊠ | 9 | 9 |
| Ethiopia | ⊠ | ⊠ | ⊠ | ⊠ | – | – | – | – | ⊠ | 5 | 5 |
| Ghana[2] | – | – | – | – | – | – | – | – | – | 0 | 0 |
| India | ⊠ | – | ⊠ | – | – | ✓ | – | – | ✓ | 4 | 2 |
| Indonesia SIGA | ⊠ | ⊠ | ⊠ | ⊠ | ⊠ | ⊠ | – | – | – | 6 | 6 |
| Kenya | ✓ | ✓ | ✓ | ✓ | ✓ | ✓ | ✓ | ✓ | ✓ | 9 | 0 |
| Malawi | ⊠ | ✓ | ⊠ | ✓ | ✓ | ✓ | ✓ | ✓ | ✓ | 9 | 2 |
| Mozambique | – | – | ⊠ | – | – | – | – | – | ⊠ | 2 | 2 |
| Nepal[2] | ⊠ | ⊠ | ⊠ | ⊠ | – | ⊠ | – | – | – | 5 | 5 |
| Nigeria | ⊠ | ⊠ | ✓ | ✓ | ✓ | ✓ | – | ✓ | ✓ | 8 | 2 |
| Pakistan (≥1) | ⊠ | ⊠ | ⊠ | ⊠ | ⊠ | ⊠ | – | – | – | 6 | 6 |
| Balochistan | ⊠ | ⊠ | ⊠ | ⊠ | ⊠ | ⊠ | – | – | – | Missing | 6 |
| KP | ✓ | ✓ | ✓ | ✓ | ✓ | ✓ | – | – | – | 6 | Missing |
| Punjab | ⊠ | – | – | – | – | – | – | – | – | Missing | 1 |
| Sindh | ✓ | ✓ | ✓ | ✓ | ✓ | ✓ | – | – | – | 6 | Missing |
| Philippines[2] | ⊠ | – | – | – | – | – | – | – | – | 1 | 1 |
| Rwanda | ⊠ | ⊠ | ✓ | ⊠ | ⊠ | ✓ | – | ⊠ | ⊠ | 8 | 6 |
| Sierra Leone | ⊠ | ⊠ | ⊠ | ⊠ | ⊠ | ⊠ | ✓ | – | ✓ | 8 | 6 |
| Tanzania | ✓ | ⊠ | ✓ | ✓ | ✓ | ✓ | – | – | – | 6 | 1 |
| Uganda | ⊠ | ⊠ | ⊠ | ⊠ | ✓ | ⊠ | – | ⊠ | ✓ | 8 | 6 |
| Zambia | ✓ | ✓ | ✓ | ✓ | ✓ | ✓ | ✓ | ✓ | ✓ | 9 | 0 |
| Total-Registers | 16 | 14 | 16 | 13 | 11 | 13 | 5 | 7 | 11 | – | – |
| Total-Summary forms | 13 | 10 | 11 | 8 | 5 | 6 | 1 | 3 | 4 | – | – |

[1]Injectable only appropriate for non-breastfeeding women within 48 hours

[2]Register(s) has instructions to write in method without a list of methods that can be recorded. We did not assume facility staff write in methods that are not in summary forms, though it is possible they write in additional methods.

[3]In some cases, registers include specific other methods (e.g., diaphragm) and in other cases there is a general "other" category.

in Burkina Faso, Ghana, and the Philippines allow providers to write in methods without specifying which ones are permitted; we assumed methods are not recorded unless they appear in the summary form.

Reporting further limits visibility into method mix. Some countries that capture multiple methods in registers do not disaggregate them in reporting forms. For example, summary forms in Kenya and Zambia do not disaggregate by method at all, while Malawi, Nigeria, and Tanzania disaggregate only one or two methods (IUDs and sterilization in Malawi, IUDs and implants in Nigeria, implants in Tanzania) with all others grouped as "other" or not reported.

## Postabortion family planning

The numerator for the HIP-recommended indicator on PAC clients counseled on FP is captured in registers in 7 countries but included in monthly summary forms in only Ghana and Pakistan's Balochistan Province (Table 3). When it comes to the HIP-recommended indicator on PAFP method provision, 14 countries capture the numerator in registers and 10 countries include it in their summary forms (Table 4).

The HIP brief also recommends disaggregating the PAFP provision indicator by age and by method. Of 14 countries where PAFP provision is recorded, 14 capture information on age in registers and 2 (Ghana and Tanzania) disaggregate

**Table 6. Capturing and disaggregating PAFP method across 18 countries.**

| Country | IUD | Implant | Steril-ization | Pill | Con-dom | Inject-able | EC | Natural Method | Other | # of methods (Registers) | # of methods (Summary forms) |
|---|---|---|---|---|---|---|---|---|---|---|---|
| Bangladesh DGFP | n/a | n/a | n/a | n/a | n/a | n/a | n/a | n/a | n/a | | |
| Burkina Faso[1] | ⊠ | ⊠ | ⊠ | ⊠ | ⊠ | ⊠ | ⊠ | ⊠ | ⊠ | 9 | 9 |
| Ethiopia | ✓ | ✓ | ✓ | ✓ | – | ✓ | – | – | ✓ | 6 | 0 |
| Ghana | ⊠ | ⊠ | ⊠ | ⊠ | ⊠ | ⊠ | – | ⊠ | – | 7 | 7 |
| India | ⊠ | – | ⊠ | ✓ | ✓ | ✓ | – | – | – | 5 | 2 |
| Indonesia SIGA | ⊠ | ⊠ | ⊠ | ⊠ | ⊠ | ⊠ | – | – | – | 6 | 6 |
| Kenya | ✓ | ✓ | ✓ | ✓ | ✓ | ✓ | ✓ | ✓ | – | 8 | 0 |
| Malawi | ⊠ | ⊠ | ⊠ | ⊠ | ⊠ | ⊠ | ⊠ | – | – | 7 | 7 |
| Mozambique | ✓ | ✓ | – | ✓ | – | ✓ | ✓ | – | – | 5 | 0 |
| Nepal | ✓ | ✓ | ✓ | ✓ | ✓ | ✓ | – | – | ✓ | 7 | 0 |
| Nigeria | ✓ | ✓ | ✓ | ✓ | ✓ | ✓ | – | ✓ | – | 7 | 0 |
| Pakistan (≥1) | ⊠ | ⊠ | ⊠ | ⊠ | ⊠ | ⊠ | – | – | – | 6 | 6 |
| Balochistan | ⊠ | ⊠ | ⊠ | ⊠ | ⊠ | ⊠ | – | – | – | Missing | 6 |
| KP | ✓ | ✓ | ✓ | ✓ | ✓ | ✓ | – | – | – | 6 | Missing |
| Punjab | n/a | n/a | n/a | n/a | n/a | n/a | n/a | n/a | n/a | | |
| Sindh | ✓ | ✓ | ✓ | ✓ | ✓ | ✓ | – | – | – | 6 | Missing |
| Philippines | n/a | n/a | n/a | n/a | n/a | n/a | n/a | n/a | n/a | | |
| Rwanda | n/a | n/a | n/a | n/a | n/a | n/a | n/a | n/a | n/a | | |
| Sierra Leone | n/a | n/a | n/a | n/a | n/a | n/a | n/a | n/a | n/a | | |
| Tanzania | ✓ | ⊠ | ✓ | ✓ | ✓ | ✓ | – | – | ⊠ | 7 | 2 |
| Uganda | ⊠ | ⊠ | ⊠ | ⊠ | ✓ | ⊠ | – | – | ✓ | 7 | 5 |
| Zambia | ✓ | ✓ | ✓ | ✓ | ✓ | ✓ | – | – | – | 6 | 0 |
| Total-Registers | **14** | **13** | **13** | **14** | **12** | **14** | **4** | **4** | **5** | – | – |
| Total–Summary forms | **7** | **7** | **7** | **6** | **5** | **6** | **2** | **2** | **2** | – | – |

[1]Service providers write in method with no code list in register, so assume any method can be recorded

n/a = Not applicable because PAFP method provision is not collected

PAFP provision by age categories in summary forms (results not shown). When it comes to disaggregation by method, 6 countries record most or all methods that could be provided to PAC clients in registers but do not disaggregate by method in summary forms (Ethiopia, Kenya, Mozambique, Nepal, Nigeria, Zambia) (Table 6). Only PAC clients receiving IUD or sterilization are included in the summary form in India, and Tanzania's summary form only disaggregates by implant while combining PAC clients receiving all other methods. The remaining 6 countries (Burkina Faso, Ghana, Indonesia, Malawi, Pakistan, Uganda) record 6 or more methods, and PAFP provision is disaggregated by method in summary forms (with the exclusion of condoms and "other" methods in Uganda, which are captured in registers but not reported).

## Discussion

This review analyzed PPFP and PAFP content in facility registers for ANC, L&D, PNC, FP, and PAC and monthly summary forms from 18 countries. It provides new information not captured in previous assessments, including a 2020 review of registers from 21 countries across multiple regions, which assessed the availability of maternal and newborn health indicators and included FP counseling but not provision [18]. That review found limited documentation of FP counseling, particularly immediately after childbirth: only 5 of 20 ANC registers, 3 of 18 L&D registers, and 8 of 13 PNC registers included it. Our review, conducted 4 years later and focused on low- and lower-middle-income countries [22], found a

higher proportion of countries documenting FP counseling, particularly around the time of childbirth: 8 of 18 document FP counseling during pregnancy, 7 of 18 document counseling prior to discharge after childbirth, and 10 of 18 countries document counseling during PNC or within 6 weeks after childbirth.

Our findings also update and expand upon a 2018 review of tools from 18 countries, 12 of which are included in this analysis [14]. Since 2018, most countries in our review have updated some or all of their HIS tools. Given their national commitments to PPFP and PAFP, changes in documentation were expected. Compared to the 2018 review, more countries document FP counseling (8 vs 5 during ANC; 7 vs 2 before discharge; and 10 vs 5 during PNC), though inclusion in summary forms remains limited. The 2018 review found only 12 of 18 countries document PPFP provision (9 in summary forms) and 7 of 18 document PAFP provision (2 in summary forms). Our review found all 18 countries document and report PPFP provision and 14 of 18 document PAFP provision, with 10 including it in summary forms.

Our analysis found that countries are more likely to report method provision in summary forms than counseling, aligning with recommendations from the PPFP Community of Practice Measurement Committee. While the HIP brief does not prioritize one indicator over the other, guidance may be helpful for countries unable to include both. Importantly, documenting counseling in registers can act as a service delivery prompt for providers [23,24], but offers limited insight into the quality of counseling [25–27]. Where counseling information is aggregated, occasional quality assessments through client interviews or observation would help interpret data.

The HIP IPPFP brief recommends tracking both counseling and method provision before discharge following childbirth [7]. While all countries in this review collect PPFP provision data, some use alternative timeframes consistent with WHO's guidance to integrate FP throughout ANC, childbirth, PNC, and infant immunization services [6]. From a measurement standpoint, IPPFP is attractive because it provides a clear denominator (facility births), enabling comparability across geographies and time. Increased facility births and low postnatal care attendance further justify this indicator [28]. Most countries in our review (15/18) are collecting IPPFP, though 6 of these need to add it to their summary forms for data to be reported upward and thus readily available for analysis. Many countries have also demonstrated a willingness to track PPFP across multiple contact points, reflecting a commitment to measuring broader PPFP integration, which could become more relevant with emerging evidence of cumulative benefits of integration [29].

Offering a range of contraceptive methods is central to rights-based FP. The PPFP Community of Practice recommended disaggregating IPPFP by method and including all WHO-approved methods, including LAM [30]. While many countries comply with this recommendation, others only capture certain methods, which can result in undercounting and reinforce misconceptions about method safety and effectiveness for postpartum use.

The committee also recommended age disaggregation for countries with high adolescent pregnancy, suggesting a simple <20/20 + breakdown to minimize reporting burden. However, the HIP briefs did not specify when age disaggregation should be applied [7,9]. Although adolescent pregnancy rates are highest in LMICs, very few countries in this review currently apply age-disaggregation to PPFP or PAFP indicators [31]. This aligns with findings from another review, which found that most countries only break down some FP data elements by age [32]. FP2020 has promoted age-disaggregated data, and countries have expressed willingness to extend the number of age-disaggregated data elements [32,33]. Still, decisions about age-disaggregation are likely to depend on individual county priorities and capacities. (See Box 2 for a country case study.) Documentation from countries that do disaggregate PPFP and PAFP by age could help inform future efforts and clarify trade-offs between data detail and reporting burden.

Even when countries report the same indicator, the choice of register, and whether data are spread across multiple registers, can affect data quality and comparability. Several participants in the FP2030 Nepal workshop reported undercounting or overcounting as problems during discussions on data availability and quality. If the register used to record PPFP is not accessible where services are offered, there may be undercounting. For example, data may be missed if PPFP is recorded in an FP register housed separately from ANC, L&D, or PNC services. Conversely, if PPFP is documented in multiple registers, underreporting can happen if only one register is used for compiling data on PPFP services provided,

while overcounting can occur if the same woman is captured in more than one register. To avoid double counting, Ethiopia includes this guidance in both the L&D and PNC registers: *"In order to avoid duplication IPPFP should be registered only if service is provided in the unit."* Similar clear instructions were not found in registers from all countries where PPFP is recorded in more than one place; countries without instructions should consider adding to prevent confusion and improve data accuracy.

### Limitations

This review focused mainly on a select group of countries that participated in the FP2030 Nepal workshop, limiting generalizability to other LMICs. We requested the most current tools; however, rollout of updated tools is often gradual, and the versions reviewed may not reflect those currently in use across all parts of the country. When data elements were unclear, we attempted to consult in-country data experts, but some data elements may have been missed or misinterpreted, particularly in tools that were not in English. While findings were shared with workshop participants, they were not verified with national HIS experts outside of that group, except for the five countries reviewed by R4S/SMART-HIPs, where findings were confirmed through direct conversations with targeted country HIS experts.

   This analysis focused on the availability of numerator data for HIP indicators and did not assess: denominator data (the number of women giving birth or PAC clients, which were presumed to be universally available in countries collecting PPFP or PAFP data), age disaggregation of counseling data as few countries report on counseling, registers for services outside the maternal continuum (e.g., immunization, nutrition, well- or sick-child visits, or general outpatient/ inpatient care), or electronic databases used for data storage and analysis [34]. Data quality and use were also not evaluated, although both are recognized challenges in LMIC routine health systems [35].

### Conclusions

This review finds that many LMICs are collecting data on PPFP provision and, to a slightly lesser extent, PAFP provision. While there is variation across countries, a sufficient number are collecting comparable data, particularly on IPPFP and PAFP, to open the possibility of meaningful cross-country learning and analysis. At the same time, many countries are tracking PPFP across multiple contact points, demonstrating a willingness to measure broader PPFP integration and likely reflecting national commitments and integration policies. National health information systems need to generate data to drive decision-making in support of national priorities. Thus, it would be useful for HIP briefs and other global guidance to include a curated menu of PPFP indicators beyond the universal recommendations around IPPFP and PAFP indicators for countries seeking to integrate PPFP more fully across the postpartum period.

> ### Box 2. Case Study: Selecting Postpartum Family Planning (PPFP) indicators in Uganda
>
> Uganda's approach to postpartum family planning (PPFP) data collection within its health information systems (HIS) reflects evolving policy priorities and evidence-informed decision-making. While integration of family planning (FP) with maternal and child health services, including postnatal and postpartum care, was acknowledged in the country's first Costed Implementation Plan for Family Planning (FP-CIP) for 2015–2020 [36], significantly greater emphasis was placed on PPFP in the second FP-CIP (2020/21–2024/25) [37]. This shift was informed by findings from a locally conducted study of PPFP services delivered through a maternal voucher scheme [38].
>
> The second FP-CIP outlines a comprehensive approach to PPFP integration across the continuum of care. Specifically, the plan calls for the integration of PPFP counseling into antenatal care (ANC) and FP service provision into postnatal care (PNC), immunization visits, and other routine health services provided during the first 12 months postpartum. New PPFP indicators were developed as part of a consultative process led by the Reproductive and Child

Health Division within the Uganda Ministry of Health (MOH). Stakeholder meetings were convened to identify emerging data needs. These needs were consolidated into a harmonized list submitted to a designated sub-committee within the government responsible for data governance, which reviewed, adopted, or rejected proposals and documented the justification.

A key outcome of this process was the decision to collect and report PPFP uptake across five distinct time intervals following childbirth: (1) within 48 hours, (2) 48 hours to 4 weeks, (3) 4–6 weeks, (4) 6 weeks to 6 months, and (5) 6–12 months postpartum. This granular reporting aimed to respond to stakeholder interest in identifying when women initiate PPFP services across the extended postpartum period.

Data are also disaggregated by contraceptive method—namely intrauterine device (IUD), implant, sterilization, oral contraceptive pills, injectable, and lactational amenorrhea method (LAM). However, with a growing emphasis on tracking equity in FP service utilization, programmatic priorities shifted to disaggregation by age group. As a result, a revised recommendation has since been adopted: PPFP data reported from service delivery points would henceforth be disaggregated by timing of uptake and age, rather than timing and contraceptive method. The reporting form is to be updated accordingly.

This forthcoming adjustment illustrates the practical trade-offs involved in HIS indicator selection. Although contraceptive method information remains available in facility registers for local use, national-level reporting deprioritized this disaggregation due to concerns about the reporting burden and associated costs.

Uganda's experience demonstrates how indicator selection within HIS is shaped by broader policy priorities, resource considerations, and evolving programmatic goals. The MOH's decision to prioritize age disaggregation over method type exemplifies the need for alignment between data collection and actionable program insights—especially in contexts where resource constraints necessitate strategic compromises.

## Acknowledgments

We acknowledge the following individuals for their guidance in gathering and interpreting tools and findings of this HIS review: Sarah Nabukeera, Samuel Nwaokomah, Farouk Issa, Funmilola OlaOlorun, Sushma Rajbanshi, Joaquim Vilanculos, Valério Govo, Maurice Sawadogo, and Track20 and Ministry of Health officials. Thank you to country delegations at the FP2030 Nepal workshop who provided information and validated results. Thank you to Jason Bremner, Emily Sonneveldt, Althea Wolfe, and Trinity Zan for including this review in the FP2030 Nepal workshop and HIP measurement webinar and for reviewing and providing feedback on the presentations. Thank you to Anne Pfitzer for supporting the original conceptualization of this endeavor and providing comments on a draft manuscript.

## Author contributions

**Conceptualization:** Deborah Sitrin, Aurélie Brunie, Rebecca Rosenberg, Lucy Wilson, Elena Lebetkin.

**Data curation:** Deborah Sitrin, Aurélie Brunie, Rebecca Rosenberg, Lucy Wilson, Elena Lebetkin.

**Formal analysis:** Deborah Sitrin, Aurélie Brunie, Rebecca Rosenberg, Lucy Wilson, Elena Lebetkin.

**Funding acquisition:** Deborah Sitrin, Aurélie Brunie.

**Investigation:** Deborah Sitrin, Aurélie Brunie.

**Methodology:** Deborah Sitrin, Aurélie Brunie.

**Project administration:** Deborah Sitrin, Aurélie Brunie.

**Resources:** Deborah Sitrin, Aurélie Brunie.

**Software:** Deborah Sitrin, Aurélie Brunie.

**Supervision:** Deborah Sitrin, Aurélie Brunie.

**Validation:** Deborah Sitrin, Aurélie Brunie.

**Visualization:** Deborah Sitrin, Aurélie Brunie, Rebecca Rosenberg, Lucy Wilson, Elena Lebetkin.

**Writing – original draft:** Deborah Sitrin, Aurélie Brunie, Rogers Kagimu, Fredrick Makumbi.

**Writing – review & editing:** Deborah Sitrin, Aurélie Brunie, Rebecca Rosenberg, Lucy Wilson, Elena Lebetkin, Rogers Kagimu, Fredrick Makumbi.

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
