## [Decision Letter · Decision Letter 0]

13 Jun 2025

PGPH-D-25-01121

Assessing the availability and scope of routine data on post-pregnancy family planning: a cross-sectional review of registers and reporting tools in 18 low- and middle-income countries

Dear Dr. Sitrin,

Thank you for submitting your manuscript to PLOS Global Public Health. After careful consideration, we feel that it has merit but does not fully meet PLOS Global Public Health’s publication criteria as it currently stands. Therefore, we invite you to submit a revised version of the manuscript that addresses the points raised during the review process.

We look forward to receiving your revised manuscript.

Kind regards,

Orvalho Augusto, MD, MPH, PhD

Academic Editor

Journal Requirements:

Additional Editor Comments (if provided):

Reviewers' comments:

Reviewer's Responses to Questions

**Comments to the Author**

1. Does this manuscript meet PLOS Global Public Health’s publication criteria ? Is the manuscript technically sound, and do the data support the conclusions? The manuscript must describe methodologically and ethically rigorous research with conclusions that are appropriately drawn based on the data presented.

Reviewer #1: Yes

Reviewer #2: Partly

2. Has the statistical analysis been performed appropriately and rigorously?

Reviewer #1: Yes

Reviewer #2: N/A

3. Have the authors made all data underlying the findings in their manuscript fully available (please refer to the Data Availability Statement at the start of the manuscript PDF file)?

Reviewer #1: Yes

Reviewer #2: Yes

4. Is the manuscript presented in an intelligible fashion and written in standard English?

Reviewer #1: Yes

Reviewer #2: Yes

5. Review Comments to the Author

Reviewer #1: Manuscript PGPH-D-25-01121: Assessing the availability and scope of routine data on post-pregnancy family planning: a cross-sectional review of registers and reporting tools in 18 low- and middle-income countries

General comments - The manuscript provides a good recent analysis of the status on availability of key PPFP data elements that are included in registers and monthly summary forms used for routine data collection in select LMICs.

Specific Comments

Abstract – The abstract is well written and summarizes the rationale for the study, methodology, key findings, and conclusions of the study.

Introduction - This section provides relevant background and rationale for the review. The authors have cited relevant indicators, initiatives geared at advancing availability or PPFP data and timelines, focusing on the focus of the manuscript.

Methods –General: While the description of the process followed to obtain the tools and reporting form from the various countries has been described well, additional details would be helpful for anyone planning to conduct a similar review in another or related health area. The authors should consider adding:

- Ensure all the acronyms are defined as some readers may not be familiar with them, e.g. MOMENTUM, R4S/SMART-HIPs

- Some registers might have been availed in electronic format (hopefully the majority). Include how tools/registers available in paper format ONLY were they handled (e.g. scanned, translated, variables typed by investigator in Ms Excell/Word document…etc.).

- Provide some more details on the analysis process, e.g. if any categories of data elements were generated, harmonization of “meanings” if different terms were used across different countries, how data were summarized and presented, etc.

- Text in rows 170-173 seem best suited for the “methods” section explaining how analysis, presentation and interpretation of findings was done; move from the results section.

Results – Findings presented well in logical subheadings making it easy to follow.

Discussion – This section can be improved/ strengthen by adding evidence based on use of longitudinal registers as opposed to cross-sectional registers to avoid overcounting / duplication (Refer to lines 299-304), especially if a woman receives multiple services in the same health facility.

References – Relevant and recent.

Reviewer #2: I - This is an interesting piece of work on postnatal care data systems in selected LMICs, which is an area which remains underreported. There is a lot of work that has gone into the analysis of the tools, which is a merit of the manuscript. However, I have some concerns over the methodology and write-up, which require attention from the Authors. This is mainly on differentiating the work done as part of this study from previous work conducted, including stronger rationale for conducting the work given the previous reviews and the fact that some of the tools have not changed in the interim period. Second main concern is over the methodology and limitations, which although acknowledged, do not seem to have been taken into account when setting up the study and therefore limiting the robustness of the work. Strengthening the discussion and recommendations for researchers, practitioners and policy-makers requires attention to ensure those are more specific and underpinned by evidence from the work conducted. Below are more details:

1. Firstly, in terms of the premise of the work covered:

- The focus is on tools covering registers and reporting tools, but there is limited insight into how those reflect the national policy. Specifically, Indicators for FP an PNC - what is the national guidance on provision and reporting? This may help to explain why indicators are or are not present and if present why they are phrased in a particular manner. This requires investigation.

- Similarly in terms of reporting, requirements at national and regional level, facility levels for registers and additional reporting tools are of relevance to the topic, yet the nuances not captured here. This needs attention.

- In addition to all the mentioned registers and reporting tools, there are patient-held cards that also include information on PNC FP and related services - this does not feature as a point in the manuscript, but may need some attention.

- Indicators: this is a point that may need expanding to make this work more easily translatable into practice - based on your review, what indicators are used that show good practice? What are the gaps? (some of this features in the discussion but is fairly broad and not supported by evidence so needs expanding). In terms of reporting - again, how does your review help to see what works and what needs improving?

- Denominators: this is mentioned as a limitation, but really is an integral part of indicators and usability of data and therefore requires insights from the Authors.

2. Secondly, there are aspects of the methods which potentially limit the usability of the work and require addressing and possible amending as part of revisions:

- In particular, the manuscript requires clear differentiation between what has been done, esp in light of dates of publication of tools used also in this review - how much of this work is duplication, how much is new needs to be more clearly delineated to show evidence of contribution to knowledge

- Limitations: The limitations presented raise serious concern over the findings and their value for reporting. Is there a reason none of the identified issues were addressed?

3. Discussion and recommendations need strengthening. At present while immediate references to other work are presented, as already mentioned the value added of this study is not sufficiently convincing. Additionally, the practical aspect of what this study means in terms of its use by a wider community of policy-makers, practitioners and researchers the statements lack underpinning by evidence from the work conducted and possibly wider literature. These sections need strengthening and sharpening.

II - Specific feedback:

- L51-56: Many off the tools reviewed here date pre 2018, therefore it is not surprising that the indicators are still missing in registers

- Table 1: Source / Reference to be included.

- L109-113: A clear statement on how this work differs from what has been done previously is necessary, especially to flag value added of this study.

- L130-131: Why not?

- L133-135: Why not?

- L287-288: Is this recommendation valid in those countries, ie. is there a high rate of teenage pregnancies that would warrant this split being used? Please expand on this.

- L297-301: What is the insight from the countries regarding these points? Would it be possible to gather some inputs form the partners to provide more depth to the argument? Or seek more insight form wider literature?

- L301-304: This is a fair point, but needs to be substantiated to make it more robust. Are you suggesting the other countries do not have this sort of guidance?

- L307-308: This makes the findings somewhat shaky!

- L308-309: Again, this is a serious issue when reporting!

- L309-312: Once more this is really big limitation!

- L312-316: As is this!

- L323-325: This is a suggestion that may require more insight - what is the current state of affairs in terms of standardisation of indicators used based on your review? What would wider literature? Please explain to provide more depth.

- Box 2: Were all countries studied in a similar manner to understand issues related to PPFP reporting? This is the level of detail required to understand the topic better.

III - Formatting, typographical errors and language:

- Author list: please check all names are separated by a comma and amend the font to make it consistent;

- All abbreviations need to be introduced at first use – please check and amend where needed (e.g. see LAM in Box 1);

- Data: please standardise is singular or plus verb form is to be used and amend as required;

- L77: ‘during’ not ‘curing’ – please amend;

- The term 'delivery' has been updated to 'childbirth' - please consider revising to meet the current standards;

- Tenses: consistency required (e.g. see L122-123);

- Table 2: formatting of the table to be checked for consistency.

6. PLOS authors have the option to publish the peer review history of their article (what does this mean? ). If published, this will include your full peer review and any attached files.

**Do you want your identity to be public for this peer review?** For information about this choice, including consent withdrawal, please see our Privacy Policy .

Reviewer #1: No

Reviewer #2: No

---

## [Decision Letter · Decision Letter 1]

31 Aug 2025

Assessing the availability and scope of routine data on post-pregnancy family planning: a cross-sectional review of registers and reporting tools in 18 low- and middle-income countries

PGPH-D-25-01121R1

Dear Ms Sitrin,

We are pleased to inform you that your manuscript 'Assessing the availability and scope of routine data on post-pregnancy family planning: a cross-sectional review of registers and reporting tools in 18 low- and middle-income countries' has been provisionally accepted for publication in PLOS Global Public Health.

The reviewer also noted some typos, which you can address during final formatting checks.

Best regards,

Julia Robinson

Executive Editor

Reviewer #2:

Reviewer Comments (if any, and for reference):

Reviewer's Responses to Questions

**Comments to the Author**

Reviewer #2: All comments have been addressed

publication criteria?

Reviewer #2: Yes

3. Has the statistical analysis been performed appropriately and rigorously?

Reviewer #2: N/A

4. Have the authors made all data underlying the findings in their manuscript fully available (please refer to the Data Availability Statement at the start of the manuscript PDF file)?

Reviewer #2: Yes

5. Is the manuscript presented in an intelligible fashion and written in standard English?

Reviewer #2: Yes

Reviewer #2: Thank you for the revised version of the manuscript. The explanations and amendments provide the required information previously missing.

The limitations listed are still a concern, but they are addressed more clearly. More in-depth assessment would have been beneficial for strengthening the robustness of the work, including addressing the raised limitations or more specific analyses as done in the Uganda country study presented. However, I understand, these can now only be used as recommendations for future work.

Please check for instances of ‘county’ and ‘counties’ where ‘country’ and ‘countries’ is meant and amend.

I have no further comments.

**Do you want your identity to be public for this peer review?** For information about this choice, including consent withdrawal, please see our Privacy Policy

Reviewer #2: No
